# Authenticity, Involvement, and Nostalgia in Heritage Hotels in the Era of Digital Technology: A Moderated Meditation Model

**DOI:** 10.3390/ijerph19105784

**Published:** 2022-05-10

**Authors:** Ibrahim A. Elshaer, Alaa M. S. Azazz, Sameh Fayyad

**Affiliations:** 1Department of Management, College of Business Administration, King Faisal University, Al-Ahsaa 31982, Saudi Arabia; 2Hotel Studies Department, Faculty of Tourism and Hotels, Suez Canal University, Ismailia 41522, Egypt; sameh.fayyad@tourism.suez.edu.eg; 3Department of Tourism and Hospitality, Arts College, King Faisal University, Al-Ahsaa 31982, Saudi Arabia; 4Tourism Studies Department, Faculty of Tourism and Hotels, Suez Canal University, Ismailia 41522, Egypt

**Keywords:** authenticity, involvement, and nostalgia, heritage hotels, digital technology, self-congruity

## Abstract

Heritage hotels attract a large number of foreign and domestic tourists by offering luxurious experiences while also providing an opportunity to learn about the history, art, culture, tastes, traditions, and sentiments present in various eras in a region’s history. The literature on heritage lodging buildings is still sparse, limited, and lacking in well-established empirical evidence. The purpose of this study is to examine the relationships between perceived authenticity, visitor involvement, and nostalgia in heritage hotels, with self-congruity serving as a mediator and digital technology as a moderator in the tested relationships. A total of 278 guests of heritage sites participated in this study and the data were analysed by a structural equation modelling and smart PLS program. The empirical results provide evidence that perceived authenticity and visitor involvement positively impact heritage hotel nostalgia experience, this relationship was strengthened by self-congruity as a mediator and was dampened by digital technology as a moderator. Additionally, the theoretical and practical implications regarding the application of these findings to the tourism and hospitality industries are discussed.

## 1. Introduction

Retro style has become popular in different disciplines and industries, including the hotel, vintage train, and restaurant industries providing an unforgettable dining experience [1]. Therefore, many countries have taken the initiative to use their unique heritage and cultural resources to increase their place-based competitive advantage. Egypt was one of the first countries to advocate for the preservation of world heritage sites and a need for a system to protect this heritage in all countries worldwide [2]. Egypt has made great efforts to sustain some heritage buildings by adapting and reusing them as heritages hotels. Historic buildings that have been renovated or remodelled are among the most popular tourist accommodations and are increasingly being placed on the heritage tourism map [3].

According to the cognitive appraisal theory (CAT), emotions are significant drivers of heritage tourism experiences [4]. To better manage heritage tourism, it is critical to consider motivation factors and emotions (i.e., nostalgia, authenticity, and involvement) and their impact on subsequent behavioural intentions and visitor experience [4]. Researchers have affirmed the positive influences of nostalgia on heritage tourism. It is an influential motivator and leads to a boost in the marketing capacity of heritage tourism [5]. Both perceived authenticity and visitor involvement are essential variables that provoke and stimulate nostalgia [6]. Self-congruity greatly influences the visitor’s emotional experiences; the concept of visitors’ self-congruity should find more attention in heritage tourism [5]. Furthermore, using digital technologies can probably improve the overall tourism experience.

The literature on heritage lodging buildings is still sparse, limited, and lacking in a well-established, systematic approach and practical evidence [7] based on cognitive appraisal theory (CAT), push–pull, self-determination, and self-congruity theories. The current study aimed to explore the relationships between perceived authenticity; visitor involvement; and heritage hotel visitor nostalgia, with self-congruity serving as a mediating variable and digital technology as a moderator variable in these relationships.

## 2. Theoretical Background and Hypotheses Development

### 2.1. Heritage Hotel

“Heritage is our legacy from the past, what we live with today, and what we pass on to future generations”, heritage tourism is defined as travel with the primary goal of learning about a location’s culture and heritage [8]. With the great growth in tourists wishing to experience heritage tourism, heritage is defined as a “contemporary epidemic” [5]. Tourists of heritage tourism seek worthy experiences and a relationship with the precedent history or memories [1]. They desire to participate in a personal “historical heritage experience,” remember more about the history of the place, join in a recreational experience, become in touch with their heritage; they desire to live with the same local customs, art, traditions, and cultures of the heritage destination [6]. In today’s hospitality industry, heritage hotels have become widespread places of accommodation [9]. According to [10], heritage hotels have advanced with higher and more stable occupancy and drawn much attention from people with higher incomes and higher literacy levels. They attract many foreign and domestic tourists by providing an opportunity for partaking in luxurious experiences, while also learning about various aspects of a region’s history, especially those which are embodied in UNESCO “World Heritage Sites” [3]. According to [11], approximately 85 percent of the general population is estimated to be current or potential heritage tourists. Heritage hotels provide these tourists with emotional, aesthetic, personal, and symbolic value. In addition, they experience the pride of visiting a historical site and culture, more so than when visiting a commercial hotel [7]. Heritage hotels provide heritage experiences and lodging simultaneously.

Adaptive reuse of historical buildings is increasingly being endorsed as a practical approach to revitalizing historic districts in cities: it allows for conservation while also providing various social, economic, and environmental benefits [12]. This effort can help extend the life of buildings, reduce demolition waste, repurpose embodied energy, and provide economic and social benefits to local communities [13]. Many heritage hotels were ancient palaces, forts, mansions, factories, stores, castles, convents, post offices, churches, or even prisons. This may be why they appeal to history or heritage buffs and have become essential venues for introducing the cultural and historical background of the past to modern society [14]. Heritage character, hotels’ historical attributes, experience quality, prior knowledge, and perceived authenticity are the factors which positively affect tourists’ intention to visit heritage hotels [15,16]; The adaptive reuse of historic buildings as heritage hotels ultimately assists in the growth of the country’s tourism sector [17]. In Bangkok, the adaptive reuse of heritage buildings as small hotels achieved numerous benefits towards the preservation of heritage buildings’ value and offered economic and social benefits to the local community [18].

### 2.2. Self-Congruity as a Mediator in the Relationship between the Perceived Authenticity of Heritage Hotels and Heritage Hotel Visitor Involvement and Heritage Hotel Visitor Nostalgia

It is critical to know what kinds of heritage attractions can meet the demands for cultural consumption. Consumers of heritage accommodation should feel depth, authenticity, and involvement with this type of accommodation, all of which support their self-congruity in order to bring them to a state of nostalgia [19,20].

In 1688, Hofer coined the concept “nostalgia” to describe Swiss soldiers who were extraordinarily homesick and longing for home [21]. Nostalgia is no longer used to describe homesickness; it is used to describe individuals attempting to discover their past [22]. The concept of nostalgia, in recent years, has become more attuned to its uses and applications. Amidst the tremendous developments of today, there is an urgent need to feel a positive emotion toward the past because of an unfulfilling present [23,24]. Because of this contrast between a positive past and a negative present, nostalgia is widely applied in the marketing and promoting of places and products [23]. Two classifications of nostalgia have been determined. (1) Historical nostalgia arises from collective memories and expresses a desire to escape contemporary life by returning to a time in the distant past regarded as superior to the present, and (2) on the other hand, personal nostalgia arises from personal memories idealizing the personally remembered past [1,25]. In the tourism industry over the last three decades, an increasing number of people worldwide have expressed a desire to visit less complex, less volatile, and less technologically advanced environments. As a result, nostalgia products, destinations, and advertising messages are rising, such as museums, heritage hotels, historical architecture, and rural areas [26]. There is a general agreement that there is a positive relationship between ad-evoked nostalgia and attitudes toward the advertisement, brand, and purchase intention [25]. According to the push–pull theory, nostalgia can be considered an inner heritage tourism motivation [27]. The push–pull factors of tourism incentives are significant in evaluating tourist behaviour [7]. The pull factor is exemplified by the travel decisions and stimuli generated by the attractive features of a tourist location, while the push factors are a socio-psychological necessity and an invisible strength that boosts visitors to travel [8]. In the context of heritage hotels, the pull factors are, for example, heritage architecture, food service, and historic streets, while examples of push factors are visitor engagement, learning, and a sense of belonging [7].

The concept of authenticity is generally recognized as the theoretical starting point for any endeavour in the context of heritage hotels. The majority of academics have recognized the importance of authenticity in framing the activities of heritage hotels by viewing this concept as a fundamental cornerstone of operations [20]. Ref. MacCannell [28] introduced the concept of authenticity to tourism studies in order to analyse the tourist experience at historical sites. Authenticity has since gained prominence in tourism research, particularly in research concerning heritage tourism [29]. There are many subcategories of authenticity, but according to [24], constructive authenticity, which has usually been termed “perceived authenticity” and relates to how individuals perceive a tourism object to be authentic, is closely associated with heritage hotels. Feelings of alienation from modern life contribute to the appeal of heritage tourist sites marketed as “authentic” [19]. Authenticity is continually associated with a feeling of nostalgia, as individuals yearn for things from “the past” in the context of rapid urbanization and modernization [6]. Generally, the degree of satisfaction with heritage tourism relies upon tourists’ perceptions of authenticity. Therefore, the hypotheses adopted are as follows:

**H1.** *Perceived authenticity of heritage hotels positively influences heritage hotel visitor nostalgia*.

Visitor involvement is defined as the degree to which customers participate in various aspects of the consumption procedures, including the product, advertising, information search, information processing, purchase decision making, and the purchasing act [30]. Therefore, according to the self-determination theory, which explains travellers’ sources of motivation, involvement in an activity can be viewed as a predictor of customer behaviour and is considered a motivational variable, reflecting the extent of the personal relevance of the decision to the person in terms of their essential objectives, values, and self-concept [31]. In the heritage hotels context, the concept of involvement urges the individual’s physical, mental, emotional, social, or spiritual engagement to activate the nostalgic experience [24]. Based on self-determination theory, a higher degree of involvement in and awareness of a heritage destination leads to a higher level of self-congruity and nostalgia; thus, a higher level of memorable tourism and hospitality is experienced [32]. Founded on the references above, the following hypotheses are proposed:

**H2.** 
*Heritage hotel visitor involvement positively influences heritage hotel visitor nostalgia.*


Heritage tourism and hospitality entail using a destination’s personality to build its brand, understanding visitors’ perceptions of that destination and creating its distinct identity [33]. Thus, a niche marketing strategy can be used for this segment. In line with this, many studies have confirmed the relevance of self-congruity theory and the need to expand its application to the tourism sphere.

Self-congruity has been defined as “the combination or degree of alignment between the image of the product/brand and the self-concept of the consumer” [34]. According to the self-congruity theory, individuals have a variety of thoughts about themselves and act accordingly to strengthen their self-image. As a result, individuals are more likely to purchase products or services compatible with their original self-concept [35].

According to [36], self-congruity is a valid theory within the context of tourism. By relying on it, it is possible to relate self-congruity to pre-travel factors, such as visit motivation and destination selection [37], and post-purchase variables such as satisfaction, revisit intention, and recommendation intention [37]. Ref. Roy and Rabbanee [38] argued that antecedent variables such as social desirability (involvement), the need for uniqueness, and status consumption (authenticity) influence consumers’ self-congruity with a brand. Ref. Zhou et al. [5] state that self-congruity greatly influences the emotional experience (nostalgia) produced in a specific purchasing situation. Ref. Liu et al. [39] confirmed that the destination image significantly influences self-congruity. Therefore, we can argue that tourist self-congruity explains the relationship between perceived authenticity and perceived visitor involvement and nostalgia in the context of heritage hotels. Based on the above, it is hypothesized as seen in Figure 1:

**H3.** 
*Perceived authenticity of heritage hotels associates positively with visitor self-congruity.*


**H4.** 
*Heritage hotel visitor involvement associates positively with customer self-congruity.*


**H5.** 
*Visitor self-congruity associates positively with heritage hotel visitor nostalgia.*


**H6.** 
*Visitor self-congruity mediates the relationship between the perceived authenticity of heritage hotels and heritage hotel visitor nostalgia.*


**H7.** 
*Visitor self-congruity mediates the relationship between heritage hotel visitor involvement and heritage hotel visitor nostalgia.*


### 2.3. Digital Technology as a Moderator in the Relationship between Visitor Self-Congruity and Heritage Hotel Visitor Nostalgia

Digital technology is used in the cultural tourism field to enhance the visitor experience and engagement [40]. AR, VR, and holographic technologies have been successfully implemented in several subsectors of the tourism business [41]. AR allows for digital signage and content at cultural heritage sites without compromising the original architecture or landscape [42]. Utilizing VR and AR in heritage tourism can help control conflicting memories by recreating historical customs and myths [43]. The Technology Acceptance Model TAM has long been the most famous theoretical framework for researching new technology adoption. The TAM has shown that perceived ease of use and usefulness, the two cognitive beliefs that support the model, are strong antecedents of adoption. On the other hand, recent works have criticized the TAM for its lack of emotions and/or affective beliefs (Antón et al., 2013). The term nostalgia tends to be emotional to a large extent. In this vein, we can argue that digital technology may moderate the relationship between self-congruity and nostalgia. Thus, the following hypothesis was formulated:

**H8.** 
*Digital technology moderates the relationship between visitor self-congruity and heritage hotel visitor nostalgia.*


**Figure 1 ijerph-19-05784-f001:**
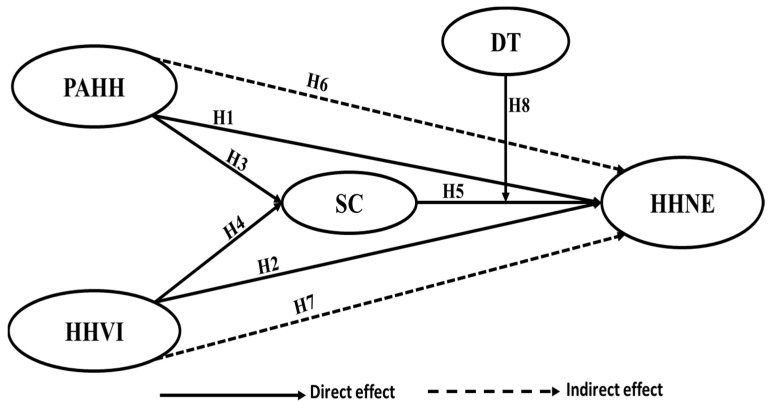
The proposed conceptual framework and hypotheses. PAHH → Perceived Authenticity of Heritage Hotels; HHVI → Heritage Hotel Visitor Involvement; SC → Self-Congruity; DT → Digital Technology; HHNE → Heritage Hotel Nostalgia Experience.

## 3. Research Methods

### 3.1. Instrument Measurement

In order to test our study’s hypotheses, a self-administrated questionnaire was developed. The scales for the study were developed after a thorough review of the literature. As a result, five dimensions were established. Ten items were used to assess the PAHH and HHVI, which were based on the findings of Rasoolimanesh et al. [8]. The SC was operationalized with a four-item scale suggested by Frias et al. [34]. The DT items were measured with a six-point scale developed by Styliani et al. and Liu [44,45]. Finally, eight items from Lee [22] were utilized to assess HHNE. The study employed a Likert scale ranging from 1 (strongly disagree) to 5 (strongly agree). The scale was tested by four academics and five practitioners for face validity and clarity; no changes were noticed.

### 3.2. Participants and Data Collection

Egypt has a large number of heritage hotels and buildings that are associated with a wide range of historical periods. By working in tourism and hotel management schools and using their connections with tourist guides at heritage sites, the research teams were able to distribute a total of 400 questionnaires. A convenient sample and drop-and-collect method was used to collect the data from guests and visitors to heritage sites (15 sites in Cairo, the Egyptian capital) during November 2021. A total of 278 valid samples were collected (the sample would be larger if there were no COVID-19), with a 69.5 percent recovery rate. The majority of the study sample—67.3 percent—were males and 32.7 percent were females aged between 26 and 71. The majority (78 percent) of participants held bachelor’s degrees, and 22% had a high school degree. A total of 60% were unmarried, while 40% were married. A full 93% of respondents were foreign guests from Germany (50%), Britain (30%), and Italy (13%), while only 7% were from Egypt.

## 4. Data Analysis Results

The current study examined the research hypotheses using “Structural Equation Modelling” (SEM) via the “Partial least squares PLS” technique with SmartPLS-3.0. The suggested and justified conceptual model was examined in two successive steps, as suggested by Leguina [46], as follows.

### 4.1. Step 1: Outer Measurement Model Assessment

To assess the reliability and validity of the outer model, several criteria were employed such as internal consistency reliability, indicator reliability, convergent validity, and discriminant validity. To begin, as shown in Table 1, the internal consistency reliability of the structures was evaluated using Cronbach’s alpha (α) values ranging from 0.785 to 0.933 and composite reliability (CR) values ranging from 0.845 to 0.948.

Second, the indicators’ degree of reliability was found to be acceptable, as all loading values were above 0.60. Third, convergent validity was determined by the presence of AVE values exceeding 0.50 [46]. Finally, three criteria were used to assess the constructs’ discriminant validity, namely, cross-loading, the Fornell–Larcker criterion, and HTMT [46]. As indicated in Table 2, the outer-loading for each latent variable (underlined) was higher than the cross-loading with other measurements.

Table 3 shows that the bolded AVEs in the diagonals exceed the correlation coefficient between variables; [47] argued that HTMT scores should not exceed 0.90. As shown in Table 3, all the HTMT readings are below 0.90, signalling a good measure of discriminant validity. The above results supported the conclusion that the study’s outer measurement model was adequate to continue evaluating the structural model.

### 4.2. Step 2: Structural Model Assessment

After obtaining a satisfactory result from the measurement outer model, the research hypotheses were evaluated by employing structural equation analysis (SQM). The model’s predictive and explanatory power were examined following the suggestion of Hair et al. [48]. The VIF scores of the observed indicators were below 5, ranging from 1.616 to 4.740, thus multicollinearity was determined not to be a problem in the structural model. Furthermore, Chin [49] has argued that the lower acceptable limit of the R2 score is 0.10. As shown in Table 4, the R2 scores of HHNE (R2 = 0.759) and SC (R2 = 0.454) were satisfactory. Moreover, The Stone-Geisser Q2 analysis showed that the HHNE and SC scores were greater than zero (Table 4), indicating a good level of predictive validity for the structural model [50].

Finally, the path coefficient and the t-values of the research hypotheses were analysed using a bootstrapping method. As shown in Table 5 and in Figure 2, PAHH was found to have a significant positive impact on both HHNE (β = 0.260, *p* < 0.01), and SC (β = 0.504, *p* < 0.01). Thus H1 and H3 were confirmed. Moreover, HHVI was found to have significant and positive impacts on both HHNE (β = 0.213, *p* < 0.01) and SC (β = 0.248, *p* < 0.003), hence confirming H2 and H4. The results signal that SC was positively associated with HHNE at β = 0.451, *p* < 0.01, confirming H4. As for the mediation effects, PAHH was found to have a positive effect on HHNE with β = 0.227, *p* < 0.01, advocating H6. Similarly, HHVI has a positive effect on HHNE via SC (indirect effect) at = 0.112, *p* 0.002, supporting H7. Finally, the results confirm that DT has a moderating effect on SC toward the HHNE, but in a negative direction at = −0.133, *p* 0.01, which contradicts H8.

## 5. Discussion and Implications

### 5.1. PAHH, HHVI, SC, and HHNE

The empirical results of this study revealed that the PAHH has a positive effect on HHNE and SC. These results are consistent with [19], who indicated that authenticity is continually associated with a feeling of nostalgia, as individuals yearn for things from “the past” in the context of rapid urbanization and modernization [6]. The results arel consistent with [39] as well, who confirmed that authenticity significantly influences self-congruity. Similarly, the results showed that the HHVI positively affected HHNE and SC. This agrees with self-determination theory, in which a higher degree of involvement in and awareness of a heritage destination leads to a higher level of self-congruity and nostalgia [32]. Furthermore, Roy and Rabbanee [38] have argued that social desirability (i.e., involvement), the need for uniqueness, and status consumption (i.e., authenticity) positively influence consumers’ self-congruity.

### 5.2. Assessing the Moderating Effect

The practical results validated the moderating influence of the DT on the relationship between SC and HHNE, but the relationship was not expected to be negative and significant. In other words, it was found that DT can dampen the positive relationship between SC and HHNE (Figure 3, Interaction plot). By returning to Figure 2 and calculating the moderator’s interaction values (0.451 + (−0.133) = 0.318), we can conclude that DT made the relationship between SC and HHNE diminish. This result disagrees with the findings of [51] but agrees with the previous criticisms made concerning TAM because of a lack of consideration for emotions and affective beliefs in its explanation for why people accept technology [52]. This may be due to those visitors who prefer to live the hospitality experience exactly as it was during the heritage hotel’s period. This kind of visitor prefers to live an authentic and not a virtual experience,’ desiring to wear the same clothes, eat the same foods in the same way, have the same service style, and experience the same celebrations as the people did who lived during the a past era. In short, he wants to live the experience of nostalgia that is in congruity with himself.

### 5.3. The Mediating Role of MTHE between the Relationship ARHH and HCL

One of the study’s main aims was to examine the mediating role of SC between PAHH and HHNE and between HHVI and HHNE. The study’s findings indicated that SC significantly mediated the relationship between PAHH and HHNE and the relationship between HHVI and HHNE. Therefore, tourist self-congruity was able to explain the relationship between perceived authenticity and perceived visitor involvement and nostalgia in the context of heritage hotels.

Based on the findings, the paper suggests that heritage hotel managers should consider the emotional aspects of visitors represented by the elements of authenticity and engagement when designing their marketing strategies to match the visitor’s self-congruity, evoking nostalgia. According to the push–pull theory, a heritage hotel can use nostalgia as a motivator to attract the kinds of visitors who have self-congruity with it; based on cognitive appraisal theory (CAT) and self-determination theory, heritage hotels can use authenticity and engagement to, through a visitor’s self-congruity, evoke nostalgia.

## 6. Limitations and Future Research Avenues

This study has some limitations as well as several suggested avenues for future research. First, the findings revealed that, in the context of heritage hotels, self-congruity played a positive mediating role in the relationships between visitors’ perceived authenticity, involvement, and nostalgia; in the same relationships, digital technology was also tested as moderator. Future research could investigate a variety of other mediating and moderating variables to gain more insight (e.g., tourist nationality, age, and gender or visiting purposes). Further research might choose to analyse not only perceived authenticity and involvement but also other aspects that influence tourists’ experiences of nostalgia in the setting of heritage hotels (i.e., previous tourist experience, culture, and social class). Moreover, in our study, a causal relationship between the examined variables cannot be firmly drawn because the data are cross-sectional. Future studies may employ longitudinal data or a mix of data sources to test this study’s model. Another consideration for future work is that the issue of the “overdoing it” might be a problem, in other words, oversaturation of the market might lead to negative consequences and a perceived lack of authenticity that might negatively affect nostalgia in heritage hotels. Finally, the suggested approach may be used to evaluate these relationships in various contexts using multigroup analysis (industry or country).

## 7. Conclusions

Many countries have taken the initiative to capitalize on their distinct cultural and heritage resources in order to strengthen their place-based competitive advantage in the global marketplace. Despite this, the research on heritage lodging buildings is still scarce and limited, and it lacks a well-established, systematic approach, as well as empirical evidence from real-world situations. Employing cognitive appraisal theory (CAT), push–pull, self-determination, and self-congruity theories, the current study examined the links between perceived authenticity, tourist involvement, and nostalgia for heritage hotels, using self-congruity as a mediator and digital technology as a moderator. During November 2021, a total of 400 surveys were gathered from guests and visitors to heritage sites using a convenient sample and drop-and-collect procedure. The hypotheses were tested using SEM with Smart PLS-3.0. The findings can be generalized to the most famous heritage houses that are acceptable for short-term living. Perceived authenticity and involvement in heritage hotels were found to have both direct impact on heritage hotel nostalgia experience and indirect impact through self-congruity, which acts as a mediating factor that strengthens the tested relationships. Nevertheless digital technology as a moderator dampened the relationships between perceived authenticity and involvement in heritage hotels and the heritage hotel nostalgia experience. This could be because the tourist wishes to experience hospitality in the manner in which it was during the hotel’s era. The visitor desires an actual experience, not a virtual one. This can be a significant challenge for heritage tourism, as guests desire to dress in period attire, eat in period restaurants, receive service in period attire, and participate in period events. In summary, they wish to live a nostalgic experience that is consistent with themselves. Additionally, the study’s findings indicate that heritage hotel managers should consider the emotional aspects of guests—as reflected by the elements of authenticity and engagement— when developing a marketing plan that corresponds to the visitor’s self-congruity, eliciting nostalgia.

## Figures and Tables

**Figure 2 ijerph-19-05784-f002:**
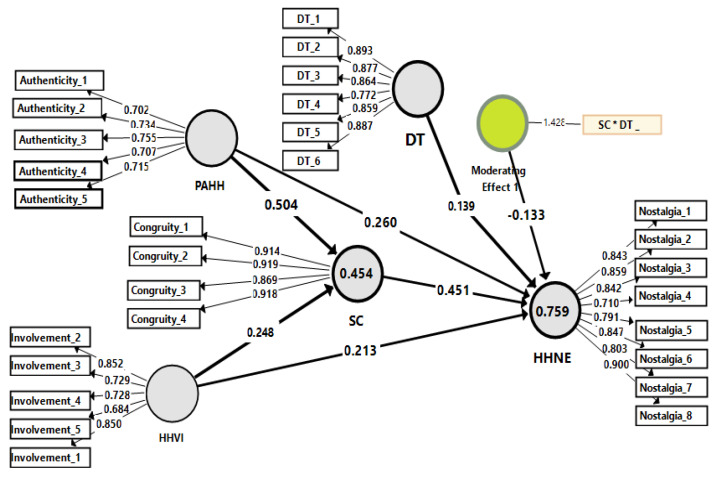
The tested structural and measurement model. R → Perceived Authenticity of Heritage Hotels; HHVI → Heritage Hotel Visitor Involvement; SC → Self-Congruity; DT → Digital Technology; HHNE → Heritage Hotel Nostalgia Experience.

**Figure 3 ijerph-19-05784-f003:**
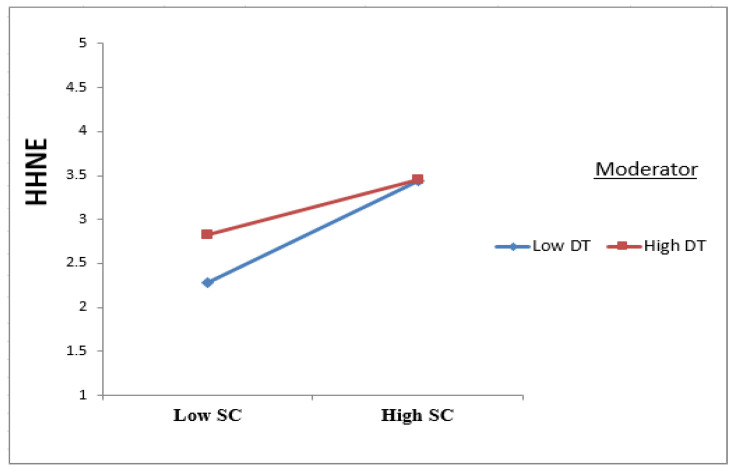
Interaction plot for the DT moderation effect on SC towards HHNE.

**Table 1 ijerph-19-05784-t001:** Evaluation of the measurement model.

Abbreviation	Outer Loading	α	C.R	AVE
PAHH		0.785	0.845	0.523
Authenticity_1	0.702			
Authenticity_2	0.734			
Authenticity_3	0.755			
Authenticity_4	0.707			
Authenticity_5	0.715			
HHVI		0.830	0.879	0.595
Involvement_1	0.850			
Involvement_2	0.852			
Involvement_3	0.729			
Involvement_4	0.728			
Involvement_5	0.684			
SC		0.926	0.948	0.819
Congruity_1	0.914			
Congruity_2	0.919			
Congruity_3	0.869			
Congruity_4	0.918			
DT		0.929	0.944	0.739
DT_1	0.893			
DT_2	0.877			
DT_3	0.864			
DT_4	0.772			
DT_5	0.859			
DT_6	0.887			
HHNE		0.933	0.945	0.682
Nostalgia_1	0.843			
Nostalgia_2	0.859			
Nostalgia_3	0.842			
Nostalgia_4	0.710			
Nostalgia_5	0.791			
Nostalgia_6	0.847			
Nostalgia_7	0.803			
Nostalgia_8	0.900			

**Table 2 ijerph-19-05784-t002:** Cross loading results.

Abbreviation	PAHH	HHVI	DT	SC	HHNE
Authenticity_1	0.702	0.626	0.249	0.577	0.626
Authenticity_2	0.734	0.601	0.201	0.602	0.577
Authenticity_3	0.755	0.196	0.496	0.330	0.407
Authenticity_4	0.707	0.087	0.406	0.282	0.443
Authenticity_5	0.715	0.245	0.504	0.373	0.417
Involvement_1	0.458	0.850	0.146	0.511	0.580
Involvement_2	0.500	0.852	0.125	0.468	0.551
Involvement_3	0.357	0.729	0.135	0.278	0.337
Involvement_4	0.460	0.728	0.214	0.345	0.485
Involvement_5	0.344	0.684	0.140	0.377	0.396
DT_1	0.447	0.176	0.893	0.295	0.347
DT_2	0.348	0.147	0.877	0.248	0.273
DT_3	0.406	0.248	0.864	0.325	0.311
DT_4	0.393	0.118	0.772	0.319	0.228
DT_5	0.429	0.165	0.859	0.236	0.291
DT_6	0.432	0.132	0.887	0.321	0.280
Congruity_1	0.581	0.489	0.324	0.914	0.795
Congruity_2	0.553	0.537	0.321	0.919	0.726
Congruity_3	0.552	0.392	0.232	0.869	0.673
Congruity_4	0.635	0.487	0.335	0.918	0.728
Nostalgia_1	0.718	0.674	0.215	0.644	0.843
Nostalgia_2	0.730	0.524	0.362	0.736	0.859
Nostalgia_3	0.517	0.583	0.268	0.625	0.842
Nostalgia_4	0.397	0.376	0.132	0.499	0.710
Nostalgia_5	0.454	0.372	0.229	0.581	0.791
Nostalgia_6	0.574	0.465	0.350	0.770	0.847
Nostalgia_7	0.533	0.516	0.311	0.596	0.803
Nostalgia_8	0.690	0.568	0.326	0.818	0.900

**Table 3 ijerph-19-05784-t003:** Inter-construct correlations, the square root of AVE, and HTMT results.

	AVEs Values					HTMT Results				
	PAHH	DT	HHNE	SC	HHVI	PAHH	DT	HHNE	SC	HHVI
PAHH	0.723									
DT	0.477	0.860				0.591				
HHNE	0.713	0.340	0.826			0.772	0.353			
SC	0.642	0.337	0.808	0.905		0.691	0.363	0.857		
HHVI	0.556	0.194	0.625	0.528	0.772	0.590	0.222	0.684	0.584	

**Table 4 ijerph-19-05784-t004:** R2 and Q2 Coefficient of determination.

Endogenous Latent Construct	(R2)	(Q2)
HHNE	0.759	0.476
SC	0.454	0.348

**Table 5 ijerph-19-05784-t005:** The structural model’s results.

	Hypotheses	Beta (β)	(T-Value)	*p* Values	Results of Hypotheses
H1	PAHH → HHNE	0.260	3.900	0.000	Accepted
H2	HHVI → HHNE	0.213	3.983	0.000	Accepted
H3	PAHH_ → SC	0.504	6.657	0.000	Accepted
H4	HHVI → SC	0.248	3.000	0.003	Accepted
H5	SC → HHNE	0.451	7.408	0.000	Accepted
H6	PAHH → SC → HHNE	0.227	4.563	0.000	Accepted
H7	HHVI → SC → HHNE	0.112	3.090	0.002	Accepted
H8	Moderating Effect 1 (SC × DT) → HHNE	−0.133	3.119	0.002	Not Accepted

## Data Availability

Data is available upon request from researchers who meet the eligibility criteria. Kindly contact the first author privately through e-mail.

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
