# Peer review of "Authenticity, Involvement, and Nostalgia in Heritage Hotels in the Era of Digital Technology: A Moderated Meditation Model"

_ijerph, 2022, doi:10.3390/ijerph19105784_

Round 1

Reviewer 1 Report

This is an interesting, innovative and concise overview of the topic, which I always see as positive writing trait. In addition to that, it's user/reader friendly, which is always a wise choice for authors.

However, I have several notes for authors:

  • The timing (Nov 21') should perhaps be contextualized within the current situation (e.g. perhaps the sample would be larger if there was no COVID-19)
  • Context matters
  • perhaps it would be wise to expand a bit on the heritage tourism, and it's consumers
  • limitations section could address or even mention the problematics of the 'overdoing it', and oversaturation of the market, that leads to negative consequences and perceived lack of authenticity that might negatively affect these processes
  • perhaps it would be wise to separate 'future research' and 'limitations' section, and address certain market need for 'amped up' inauthentic experiences one/tourist might want, regardless of the 'real and authentic' product heritage hotel might be offering. This could be even interpreted as the demand for the 'real experience' which would still be distorted version of the 'real experience' from the historic ages
  • also, perhaps would be wise to expand on push and pull- even in one sentence
  • The guests' wants to dress in period attire, eat 346
    in period fashion, get service in period fashion, and participate in period events
    this is repetitive, perhaps paraphrase in the conclusions section, also could be contextualized as a potential pitfall of the heritage tourism
  • how many heritage sites were involved?
  • perhaps a more in-depth evaluation of the 'participants and data collection' would be needed, as this section is extremely brief.

Overall, I believe this article should be published. The authors could expand on certain matters a bit more (data collection, interpretation of results, contextualization, limitations), but as a novel insight into the intersection of technology, emotional experiences and heritage tourism, I see it as a valuable contribution to the field.

Author Response

Dear Reviewer,

Thank you for giving us the opportunity to submit a revised draft of our manuscript titled “Authenticity, Involvement, and Nostalgia to Heritage Hotels in the Era of Digital Technology: A Moderated Meditation Model.” to IJERPH journal. We appreciate the time and effort that you have dedicated to providing your valuable feedback on our manuscript. We are grateful to the reviewers for his insightful comments on our paper. We have been able to incorporate changes to reflect the suggestions provided in our revised manuscript. We have colored the changes within the manuscript in red.

attached is a point-by-point response to the reviewers’ comments and concerns.

Reviewer 2 Report

The article analyzes an interesting topic from the from a hotel perspective. 
Although the effort on the research seems relevant, the paper must be reinforced. However, the paper in its current form needs the suggested improvements in order to overcome the listed limitations and produce a stronger contribution to the state of existing knowledge.

Abstract
A good abstract should keep that reader's attention and help to excite them to want to read the entire article. The current abstract is dull and if one and it seems to be (a bit) redundant. A good abstract should answer the following questions in a way that makes the prospective readers want to read the entire article.
What is the problem or challenge that the authors addressed? Why is it important/relevant/interesting? What did the authors do to provide answers/solutions to the problem or challenge? What did the authors learn? What did the authors recommend to the readers about what should be done to address the problem or challenge? Why or how will that help society/business managers to make progress?

Introduction
A reading of the introduction reveals a lack of line or continuity. Therefore, a restructuring of the introduction is recommended. On the other hand, the authors also need to present a clear research question along with the originality of this study. In the same way, the GAP of the research must be identified.

Literature review
Although the content included is interesting, this section should be strongly reinforced. The review of the literature must base its arguments on the basis of a greater number of references. Major research will enable a full comprehension of the state of the art and therefore to know the gap (to the reader)

Methodology
This section is well written. However, the writing style (very long boards) often makes the reader lost here and there. Please align with previous sections and research questions.

Conclusion
Analysing the discussions, it is necessary to consider whether this study is innovative and its real contribution. Furthermore, the inclusion of a section on practical and theoretical implications is recommended.

Author Response

(The authors gave the same response as above.)
